# Coupled strontium-calcium isotopes in Archean anorthosites reveal a late start for mantle depletion

Matilda Boyce [1] ✉, Anthony Kemp [1], Chris Fisher[1], Dan Bevan [1,2], Aleksey Sadekov[2], Jamie Lewis[3], Simon Wilde [4], Tim Ivanic[5] & Tim Elliott [3]

Voluminous felsic continental crust is, as far as we know, unique to Earth, yet the timescales of its earliest growth remain debated. Archean mantle evolution is complementary to crustal growth but is largely unconstrained for strontium isotopes due widespread Rb-Sr disturbance. Here, we perform high spatial resolution radiogenic Sr and Ca isotope measurements in magmatic plagioclase megacrysts from 3.7–2.8 Ga Archean anorthosites and leucogabbros. We report mantle-like Ca isotope signatures and the most unradiogenic terrestrial $^{87}Sr/^{86}Sr$ ratio yet measured on Earth, in plagioclase ($^{87}Sr/^{86}Sr_{initial} = 0.700050 \pm 0.000017$, 95% confidence interval) from the 3.73 Ga Manfred Complex of the Narryer Terrane, Western Australia. These data are consistent with $^{87}Sr/^{86}Sr$ homogenisation between the Earth and Moon ($^{87}Sr/^{86}Sr_{initial} \approx 0.699061$) at ca. 4.515 Ga during the Moon-forming giant impact. The appearance of a depleted mantle signature in the terrestrial strontium record post-3.5 Ga suggests that extensive continental growth began relatively late.

The timing and rate of early crustal growth on Earth remains contentious due to the scarcity of very ancient (>3.5 Ga) rocks. Long-lived radiogenic isotope systems like Nd and Hf offer invaluable insight into early crustal differentiation processes through the formation of two complementary reservoirs, the continental crust and the depleted mantle. Current models are mainly based on Hf isotope compositions of the robust accessory mineral zircon and predominantly measure growth of felsic crust[1-3]. Available evidence suggests the earliest crust was mafic in composition[4-7], however, directly measuring early mantle depletion, as a proxy for crustal growth, is challenging because Archaean mantle-derived mafic rocks are typically highly metamorphosed and do not preserve their original bulk isotopic compositions[8]. Being $SiO_2$-poor, mafic rocks are also not well represented in the zircon Hf crustal record[9]. Therefore, new approaches are required to better define the isotopic evolution of the depleted mantle. The $^{87}Rb–^{87}Sr$ isotope system has a half-life of ~49.61 billion years[10] and has long been recognised as a sensitive tracer of both crustal and planetary differentiation processes. Plagioclase feldspar is a major repository of Sr in most igneous rocks and has very low Rb/Sr, allowing for the accurate determination of initial Sr ratios in well-preserved samples[11,12]. Importantly, plagioclase dominates the Sr budget in some mantle-derived rocks, such as anorthosites.

The strontium isotope compositions of early lunar rocks and meteorites are well constrained[13-15]. However, there are few reliable data for the early Earth, largely because the Rb-Sr system is easily disturbed by post-crystallisation processes, especially in hydrous environments. Rubidium is a moderately volatile element, whereas strontium is moderately refractory. The Earth is depleted in rubidium (Rb/Sr ≈ 0.03, as estimated from the Ba/Sr of chondrites, terrestrial Rb/Ba, and isotopic compilations[15-19]) relative to carbonaceous chondrites (Rb/Sr ≈ 0.3), and its initial $^{87}Sr/^{86}Sr$ is therefore poorly constrained and depends on the timing and extent of volatile depletion during accretion. Prior to the Moon-forming giant impact, the proto-Earth may have evolved for millions of years with significantly elevated Rb/Sr[20-22].

[1]School of Earth and Oceans, The University of Western Australia, Perth, WA, Australia. [2]Centre for Microscopy, Characterisation and Analysis, The University of Western Australia, Perth, WA, Australia. [3]Bristol Isotope Group, School of Earth Sciences, University of Bristol, Bristol, UK. [4]School of Earth and Planetary Sciences, Curtin University, Bentley, WA, Australia. [5]Department of Energy, Mines, Industry Regulation and Safety, Geological Survey of Western Australia, East Perth, WA, Australia. ✉e-mail: matilda.boyce@uwa.edu.au

The most unradiogenic terrestrial $^{87}Sr/^{86}Sr$ previously measured precisely is from ~3.5 Ga barites[22], however, their composition is distinctly more radiogenic than the presumed bulk Earth at that time. This indicates that either barite incorporated some crustal Sr, or the bulk Earth has a higher initial $^{87}Sr/^{86}Sr$ or time-integrated $^{87}Rb/^{86}Sr$ than typically assumed[22,23]. However, a very high initial Earth $^{87}Sr/^{86}Sr$ is at odds with estimates of the lunar initial $^{87}Sr/^{86}Sr$, assuming the Earth and Moon equilibrated at the time of the giant impact[20]. Therefore, more accurately defining the strontium evolution of the early Earth has important implications for both crustal evolution and planetary differentiation processes.

Compared to the Rb-Sr isotope system, the radiogenic $^{40}K$-$^{40}Ca$ isotope system has a short half-life of 1.3 billion years. Magmatic processes strongly fractionate K/Ca, with the most significant radiogenic ingrowth generated by high K/Ca Archaean felsic crust. The much shorter half-life of $^{40}K$ relative to $^{87}Rb$ and the much higher concentrations of Ca in plagioclase compared to Sr means that, compared to Sr isotopes, plagioclase is more retentive of its primary Ca isotopic signature during later disturbance events[24]. The mantle, with a very low K/Ca ratio of ~0.01 and thus a near constant Ca isotope composition through time, acts as a baseline from which ancient crustal contamination of mantle-derived magmas, or major isotopic disturbances, can be detected. Measurements of $^{40}Ca/^{44}Ca$ are expressed here in parts per million as $\Delta'_{42/44}^{40}Ca_{915a}$, internally normalised to $^{42}Ca/^{44}Ca$, and externally to SRM 915a[25]. Most estimates of the $\Delta'^{40}Ca_{915a}$ composition of either the present-day or ancient mantle range between −20 and −100 ppm[25–29]. The total radiogenic ingrowth for the mantle from the Earth's formation to the present is ~12 ppm, assuming a K/Ca of 0.01, a negligible difference well within the range of estimated mantle compositions. By applying coupled radiogenic K-Ca and Rb-Sr isotope measurements to plagioclase grains, crustal contamination can be detected. Only intrusions returning mantle-like $\Delta'^{40}Ca_{915a}$ compositions are used here to define the mantle evolution for Sr isotope ratios.

Here, we apply high-precision microanalytical techniques to retrieve the Ca and Sr isotopic record of plagioclase crystals from a suite of Eoarchean to Mesoarchean anorthosites and related rocks to better constrain the evolution of the early mantle. This approach allows for the detection of crustal contamination using radiogenic Ca isotopes, while Rb-Sr disturbance can be monitored using high spatial resolution LA-MC-ICPMS micro-sampling. We show that these samples retain mantle-like Ca isotope signatures, precluding any significant contamination by felsic crust, and preserve unradiogenic initial Sr isotope ratios, including the lowest terrestrial $^{87}Sr/^{86}Sr$ ratio yet measured, recorded in plagioclase from the 3.7 Ga Manfred Complex of the Yilgarn Craton. From these results, we propose a refined Bulk Earth strontium isotope evolution model consistent with isotopic equilibration between the Earth and Moon during the terminal giant impact. The emergence of terrestrial depleted mantle signatures after ~3.5 Ga indicates that significant continental growth commenced relatively late in Earth history.

## Results and discussion

### Strontium and calcium isotopic compositions of Archaean anorthosite complexes

Archaean anorthosite complexes are a distinctive minor component of many Archaean terranes. They are cumulates formed from broadly basaltic, possibly hydrous, mantle-derived parental magmas, likely in an oceanic setting; however, a more direct analogy to modern tectonic settings is not possible[30,31]. Megacrystic calcic ($An_{80\pm10}$) plagioclase is characteristic of these complexes and is an ideal candidate for determining initial Sr and Ca ratios due to the low parent-daughter ratios for both isotope systems. Three Archaean anorthosite-bearing complexes, the 3.73 Ga Manfred Complex, Western Australia[32], the 2.94 Ga Fiskenæsset Complex, Greenland[33], and the 2.81 Ga Windimurra Igneous Complex, Western Australia[34], were selected for radiogenic Sr and Ca

isotope analysis (Sample descriptions available in Supplementary Data 1). All have mantle-like hafnium isotopic compositions[33,35,36]. Coarse (~0.5–10 cm) plagioclase crystals from anorthosites and leucogabbros that are calcic, dark grey coloured (due to fine magmatic inclusions), optically clear and preserve sharp magmatic twinning were selected for in situ Sr analysis by LA-MC-ICPMS (Supplementary Figs. 1–3). Plagioclase domains that were fractured, contained mosaic textures, or were white in colour—indicating the expulsion of primary inclusions during recrystallisation and a more albitic composition—were avoided. Several plagioclase grains with low $^{87}Sr/^{86}Sr$ were subsequently micro-milled and analysed for Sr and Ca isotopes by TIMS and CC-MC-ICPMS, respectively. Full analytical procedures are available in the Methods section.

Isotopic heterogeneity is common in Archaean magmatic rocks and may be attributed to open-system behaviour during metamorphism or alteration[37]. In situ studies of minerals such as zircon[38,39], apatite, titanite, and allanite[40–43] have been used recently to screen for open-system behaviour and to clarify the Archaean Hf and Nd isotopic records. As plagioclase is easily altered in the presence of fluids and Rb and Sr are relatively mobile, in situ techniques are particularly advantageous for successfully determining primary Archaean initial $^{87}Sr/^{86}Sr$ in the pristine portions of crystals. Samples selected for this study are well-preserved Archaean examples; however, all have experienced post-crystallisation processes of varying intensity. As radiogenic ingrowth corrections for $^{87}Sr/^{86}Sr$ are minimal in well-preserved plagioclase, re-equilibration of primary plagioclase with radiogenic fluids during metamorphism or subsequent low-temperature alteration is the primary source of potential disturbance. For each studied intrusion, the lowest age-corrected plagioclase $^{87}Sr/^{86}Sr_{initial}$, as determined by either the LA-MC-ICPMS weighted average from low $^{87}Rb/^{86}Sr$ domains or by TIMS, is therefore taken as the best estimate of the maximum mantle $^{87}Sr/^{86}Sr$ at the time of emplacement (summarised in Table 1).

All samples display a degree of present-day Sr isotope heterogeneity. Individual $^{87}Sr/^{86}Sr$ measurements by both methods range between 0.69993–0.70883 for the Manfred Complex, 0.70074–0.70187 for the Windimurra Igneous Complex, and 0.70068–0.70267 for the Fiskenæsset Complex (typical 2SE of ±0.0001 for LA-MC-ICPMS and ±0.00002 for TIMS; full results presented in Supplementary Data 2–4). Relict magmatic plagioclase was found to locally retain unradiogenic $^{87}Sr/^{86}Sr$ even at high metamorphic grades provided the plagioclase crystals were sufficiently coarse and were not entirely recrystallised or altered (e.g. Fig. 1, Supplementary Fig. 4). Hydrothermal alteration was shown to cause the most scatter in $^{87}Sr/^{86}Sr$, particularly in Manfred Complex plagioclase domains containing fine-grained secondary minerals such as epidote (e.g. Fig. 1). To isolate the freshest areas, an interpolated map of LA-MC-ICPMS point analyses was constructed across the best-preserved Manfred Complex plagioclase megacryst (Fig. 2a–d). Measurements of highly radiogenic, Rb-enriched zones are concentrated along the visibly fractured and altered (white) areas of the crystal, with low $^{87}Sr/^{86}Sr$ retained in small, optically dark plagioclase domains. From this map, the weighted mean of the two largest unradiogenic domains gave a measured $^{87}Sr/^{86}Sr$ of 0.700115 ± 17 and $^{87}Sr/^{86}Sr_{initial}$ of 0.700050 ± 17 at 3730 Ma (95% confidence interval, $n = 35$; Fig. 2e, f). This measurement represents the least radiogenic $^{87}Sr/^{86}Sr$ determined precisely in any terrestrial sample to date. The Sr isotope map was then used to guide micro-milling (Supplementary Fig. 5), although the spatial resolution of the micro-milling techniques employed meant that completely avoiding fine-grained secondary minerals or fractures at depth was difficult. This resulted in mixed analyses and more scattered results extending to higher $^{87}Sr/^{86}Sr$ when measured by TIMS ($^{87}Sr/^{86}Sr$ = 0.70033–0.70819; Supplementary Data 4). These results highlight the merits of high spatial resolution in situ analysis in complex mineral samples.

**Table 1 | Summary of strontium isotope measurements of Archaean anorthosites**

| Intrusion | Age (Ga) | Sample | An(mol%) mean | $^{87}Sr/^{86}Sr$ | $^{87}Sr/^{86}Sr_{initial}$ | Method |
|---|---|---|---|---|---|---|
| Manfred Complex, Western Australia | 3.73 | Megacrystic leucogabbro 12N8 | 79 | 0.700115 ± 17 (95% CI) | 0.700050 ± 17 (95% CI) | LA-MC-ICPMS (wt. mean) |
| Fiskenæsset Complex, Greenland | 2.94 | Megacrystic leucogabbro 151033 | 88 | 0.700816 ± 24 (2SE) | 0.700624 ± 24 (2SE) | Micro-mill TIMS |
| Windimurra Igneous Complex, Western Australia | 2.81 | Megacrystic anorthosite 21MBY04 | 84 | 0.700909 ± 9 (2SE) | 0.700829 ± 9 (2SE) | Micro-mill TIMS |

Summary of the best-certainty least radiogenic strontium isotope measurements and initial ratios determined by this study for the Manfred Complex, the Fiskenæsset Complex, and the Windimurra Igneous Complex (values plotted on Fig. 4). Complete results are available in Supplementary Data 2–4. The least radiogenic measurement for the Manfred Complex is determined from the LA-MC-ICPMS map (Fig. 2). The best-certainty least radiogenic measurements for the Fiskenæsset Complex and Windimurra Igneous Complex were determined by micro-mill TIMS. These initial ratios represent an upper limit for the mantle at the time of crystallisation.

All three intrusions recorded plagioclase initial $\Delta'^{40}Ca_{915a}$ within uncertainty of the mantle value (Fig. 3). The exceptionally well-preserved Windimurra Igneous Complex revealed a slightly elevated $\Delta'^{40}Ca_{915a}$ of $-20 \pm 15$ ppm (2SE), at the upper limit of mantle estimates. In detail, results show plagioclase $^{87}Sr/^{86}Sr_{initial}$ increasing from the lower zone ($0.700829 \pm 9$, 2SE) to the upper zone ($0.700956 \pm 20$, 2SE) of the intrusion with decreasing An# (Supplementary Data 3), possibly indicating a minor amount of crustal assimilation during emplacement. In contrast, the least altered samples from the Fiskenæsset and Manfred Complexes recorded more variable $^{87}Sr/^{86}Sr$ yet retain mantle-like $\Delta'^{40}Ca_{915a}$ of $-60 \pm 13$ and $-45 \pm 14$ ppm (2SE), respectively, indicating that any disturbance was relatively recent. As no evidence of a significant crustal component was detected in any of the three intrusions, all were included in the strontium evolution model.

## Initial strontium isotope composition of the Earth and Moon

The Earth's initial $^{87}Sr/^{86}Sr$ and bulk Sr isotope evolution are not well constrained. In addition to the lack of reliable data from ancient terrestrial rocks, part of the difficulty with estimating the Earth's Rb-Sr isotopic evolution is that the bulk Earth composition cannot be estimated from chondritic meteorites, as is the case for other isotope systems, due to the volatility of Rb[13,14,22,44,45]. Instead, the bulk Earth $^{87}Sr/^{86}Sr$ evolution is usually estimated by extrapolating modern bulk Earth estimates back to the basaltic achondrite best initial (BABI), which may or may not be representative of the Earth's initial strontium isotope composition[13,14]. Evidence from other isotope systems indicate the Earth and Moon equilibrated during a high-energy impact[46–49]. If this applies to $^{87}Sr/^{86}Sr$, lunar anorthosite data could aid in anchoring the Earth's initial ratio. However, modelled initial $^{87}Sr/^{86}Sr$ for the Earth and the Moon are inconsistent, with complicated models proposed to explain the apparent discrepancy between homogenisation of isotope systems other than Sr[15,20,22,44]. Using the newly determined lowest terrestrial $^{87}Sr/^{86}Sr$ from Manfred Complex plagioclase, we revisit whether Sr isotope equilibration between the Earth and Moon is consistent with other currently available data.

Ferroan anorthosites began to crystallise from the lunar magma ocean shortly after the Moon formed—within 1000 years, for a duration of ~10 Myr according to some models[50] – and can be used to estimate the lunar initial $^{87}Sr/^{86}Sr$. Although many ferroan anorthosite samples are heavily reworked and isotopically disturbed by repeated impacts, making isochron dating challenging, relict primary plagioclase from ferroan anorthosite 60025 returned a precise Pb–Pb age of $4.51 \pm 0.01$ Ga[51] and an unradiogenic $^{87}Sr/^{86}Sr_{initial}$ of $0.699062 \pm 1$[15]. This age is consistent with the KREEP zircon Lu-Hf model age of $4.51 \pm 0.01$ Ga[52] and is taken here to represent the primary crystallisation age of this anorthosite derived from the lunar magma ocean. Based on Hf-W constraints and lunar magma ocean crystallisation models, the age of the Moon is likely to be several million years older, assumed here to be ca. 4.515 Ga[20,50,53,54]. Using the $^{87}Sr/^{86}Sr_{initial}$ of ferroan anorthosite 60025 and an average bulk Moon $^{87}Rb/^{86}Sr$ of

$0.019$[44] gives a lunar initial $^{87}Sr/^{86}Sr$ of $0.699061 \pm 1$ at 4.515 Ga (Fig. 4c). Assuming the Earth and Moon equilibrated and share this initial ratio, and that based on near-chondritic zircon Hf measurements the Manfred Complex plagioclase $^{87}Sr/^{86}Sr_{initial}$ of $0.700050 \pm 17$ represents an upper limit for the primitive mantle at 3.73 Ga, this gives a maximum time-integrated $^{87}Rb/^{86}Sr$ of $-0.0897$ and present-day $^{87}Sr/^{86}Sr$ of 0.7049 for the bulk Earth. This value is within the loosely constrained range of modern bulk Earth estimates for $^{87}Rb/^{86}Sr$ (0.080–0.091) and $^{87}Sr/^{86}Sr$ (-0.7043–0.7050) as derived using the Sr-Nd correlation, assuming the bulk Earth is chondritic for Nd isotopes[22,55–58], in contrast to the significantly higher barite BEBI value (Bulk Earth Best Initial = 0.6994 at 4.5 Ga and modern bulk Earth = 0.7055)[22]. Our model provides an Earth initial $^{87}Sr/^{86}Sr$ estimate that both reconciles the Earth's early evolution with the currently available terrestrial and lunar data and is consistent with $^{87}Sr/^{86}Sr$ homogenisation between the Earth and Moon at the time of the giant impact.

## Strontium isotopic evolution of the Archaean depleted mantle

The least radiogenic results from this study, together with a compilation of published high-precision Archaean Sr isotope data from low Rb/Sr minerals plagioclase and clinopyroxene (Fig. 4a, b), show that a depleted mantle signature is globally apparent in the Rb-Sr system by ca. 3 Ga, with significant mantle depletion commencing at 3.5 Ga. Our results demonstrate good agreement with the seawater strontium isotope curve, which, although based on limited data for the early Archaean, indicates the composition of seawater deviated from the mantle by 3.5 Ga, marking the beginning of extensive exposure and erosion of felsic continental crust into the oceans[59–63]. We find no evidence for an early (>4 Ga) depleted mantle reservoir in the available data. This is consistent with previous Rb/Sr modelling that suggests juvenile crust production in the early Archaean was small in volume and predominantly mafic in composition, resulting in limited Rb/Sr fractionation in the Eoarchean mantle[7].

For the long-lived radiogenic isotope systems Hf, Nd, and Sr, the depleted mantle curve has traditionally been depicted as a hypothetical line projected from the modern depleted mantle back to a bulk Earth value at 4.5 Ga. Starting the mantle depletion curve at 4.5 Ga is arbitrary, however, and is not supported by the Hf and Nd data, with recent arguments made for the depleted mantle curve to instead commence at 3.8 Ga for the Hf and Nd systems[36,39]. We propose that the same be implemented for the Sr-depleted mantle curve, using 3.8 Ga as a maximum age for the onset of depleted mantle evolution.

## Methods
### Sample selection and screening
Samples of anorthosite, leucogabbro, and associated plagioclase-rich rock were collected from outcrop of the Manfred Complex, Narryer Terrane, Western Australia, and from outcrop and drillcore from the Windimurra Igneous Complex, Youanmi Terrane, Western Australia.

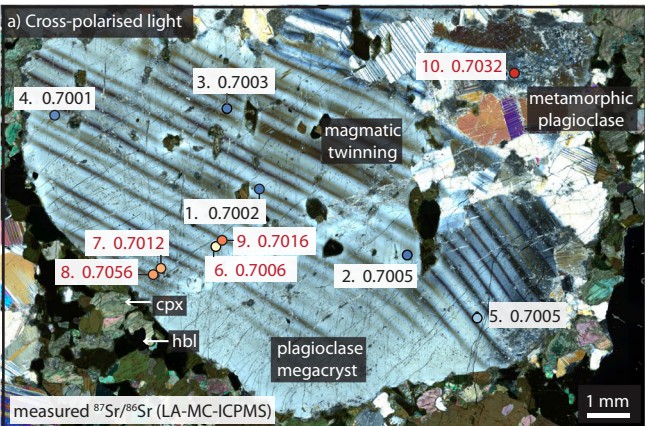

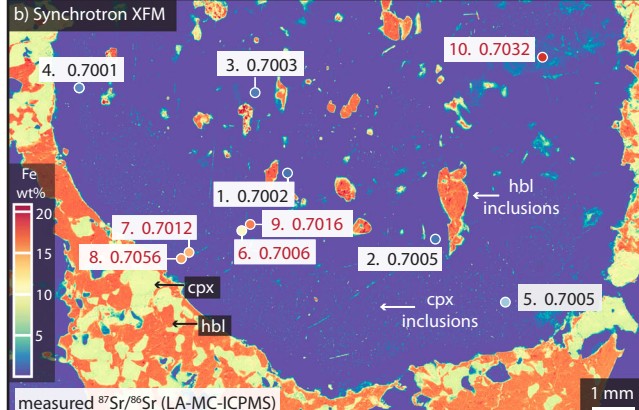

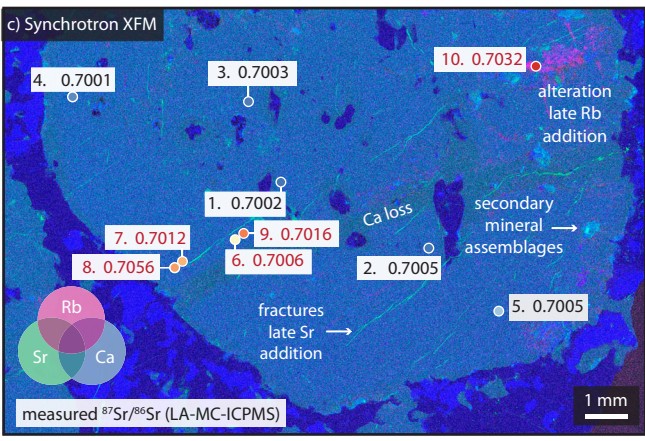

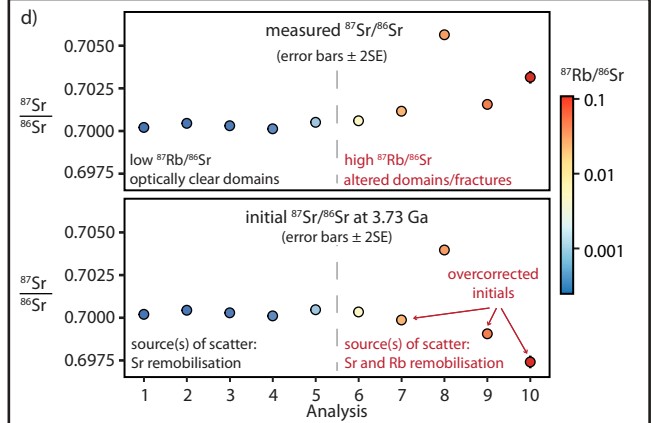

**Fig. 1 | Rb and Sr variability within plagioclase from the 3.73 Ga Manfred Complex.** Representative plagioclase megacryst from the 3.73 Ga Manfred Complex (leucogabbro 22MBN07). Localised late addition of highly mobile Rb and Sr likely occurred during the multiple thermal events that affected the Manfred Complex between ca. 3.7 and 1.8 Ga[84]. Measured LA-MC-ICPMS $^{87}Sr/^{86}Sr$ are plotted in the white boxes (no age corrections applied). Black text denotes Sr LA-MC-ICPMS spots placed on optically clear domains, and red text denotes spots placed on alteration zones or fractures. **a** Cross-polarised optical image. Magmatic twinning is visible within the plagioclase megacryst, but fractures are abundant, and some areas are partially altered and recrystallised. **b** Synchrotron X-ray fluorescence microscopy (XFM) map of Fe concentration (wt%). Fine magmatic clinopyroxene (cpx) and coarse hornblende (hbl) inclusions are abundant within plagioclase. **c** Synchrotron XFM map of Sr (green: 0–1000 ppm), Ca (blue: 0–23 wt%) and Rb (magenta: 0–520 ppm). Assemblages of fine-grained secondary minerals occur

locally. Although both Rb and Sr are visibly remobilised by secondary processes, they are distributed differently. Altered Rb-enriched domains demonstrate both highly elevated Rb/Sr and $^{87}Sr/^{86}Sr$ compared to optically clear domains (e.g. analysis 10). Fractures enriched in Sr demonstrate highly elevated $^{87}Sr/^{86}Sr$, but only moderately elevated Rb/Sr (e.g. analysis 8). **d** Plot of measured and initial $^{87}Sr/^{86}Sr$ for each LA-MC-ICPMS analysis (±2SE), with symbol fill colours representing measured $^{87}Rb/^{86}Sr$. Uncertainties are smaller than most symbols. Low $^{87}Rb/^{86}Sr$ measurements require negligible radiogenic ingrowth corrections: the main source of scatter in initial $^{87}Sr/^{86}Sr$ is due to Sr remobilisation. Measurements with high $^{87}Rb/^{86}Sr$ can result in significantly overcorrected initial $^{87}Sr/^{86}Sr$, as most of the Rb is secondary, as well as the added potential for Sr remobilisation. This demonstrates why it is crucial to only include analyses with both very low Rb/Sr and unradiogenic $^{87}Sr/^{86}Sr$ to reliably determine initial $^{87}Sr/^{86}Sr$ in Archaean materials (e.g. Fig. 2).

Samples from the Majorqap Qâva outcrop area of the Fiskenæsset Complex, Greenland, were provided by John Myers[64]. Representative hand sample photographs and optical photomicrographs are available in Supplementary Figs. 1–3. Samples were selected based on preservation of magmatic textures and mineral assemblages, minimal alteration and weathering, and low deformational strain. Thin sections were used to further identify the best-preserved plagioclase. Rock samples were cut using a diamond saw, and the visibly freshest domains were drilled out using a diamond drill press, then mounted in epoxy resin and polished.

Plagioclase mounts and thick sections were imaged optically using a Nikon petrographic microscope and were carbon-coated and imaged by BSE and CL using a Tescan Vega3 SEM at the CMCA, University of Western Australia. Typical SEM operating conditions were a working distance of 15 mm, 15 kV, 15 BI for BSE imaging and 20 kV, 20 BI for CL imaging. Plagioclase An contents (Ca/[Ca + Na + K] × 100, molar%) were determined using a JEOL 8530 F+ microprobe at the CMCA, University of Western Australia, operated using a 20 nA beam current, 15 kV voltage and a 10 μm defocused beam.

To investigate the mobility of Sr within plagioclase crystals, high-resolution microbeam μXRF mapping of select thin sections (e.g. Fig. 1) was performed on the XFM beamline at the Australian Synchrotron, operated by ANSTO. Maps were collected at an energy of 18,500 eV, with a beam size of 4 μm, a movement speed of 16 mm/s and a dwell time of 0.25 ms/pixel. The GeoPIXE software package was used to process data and produce false-colour element maps[65].

**Strontium isotopes by LA-MC-ICPMS**

In situ plagioclase strontium isotopes were acquired over multiple sessions by LA-MC-ICPMS at the CMCA, University of Western Australia, using a ThermoFisher Neptune Plus MC-ICPMS coupled with a Teledyne Photon ANALYTE G2 193 nm ArF Excimer laser system, with amplifier gains calibrated before each session. Measurements were made in low resolution, with masses 83.5 ($^{167}Er^{++}$), $^{84}Sr$, $^{85}Rb$ (axial), $^{86}Sr$, 86.5 ($^{173}Yb^{++}$), $^{87}Sr$, and $^{88}Sr$–measured on Faraday detectors with $10^{11}$ Ω amplifiers. Measurements of $^{82}Kr$ and $^{83}Kr$ beams were made using $10^{13}$ Ω amplifiers to monitor for interference by CaCa and CaAr species following the data reduction scheme of Mulder et al.[66]. Due to low Sr

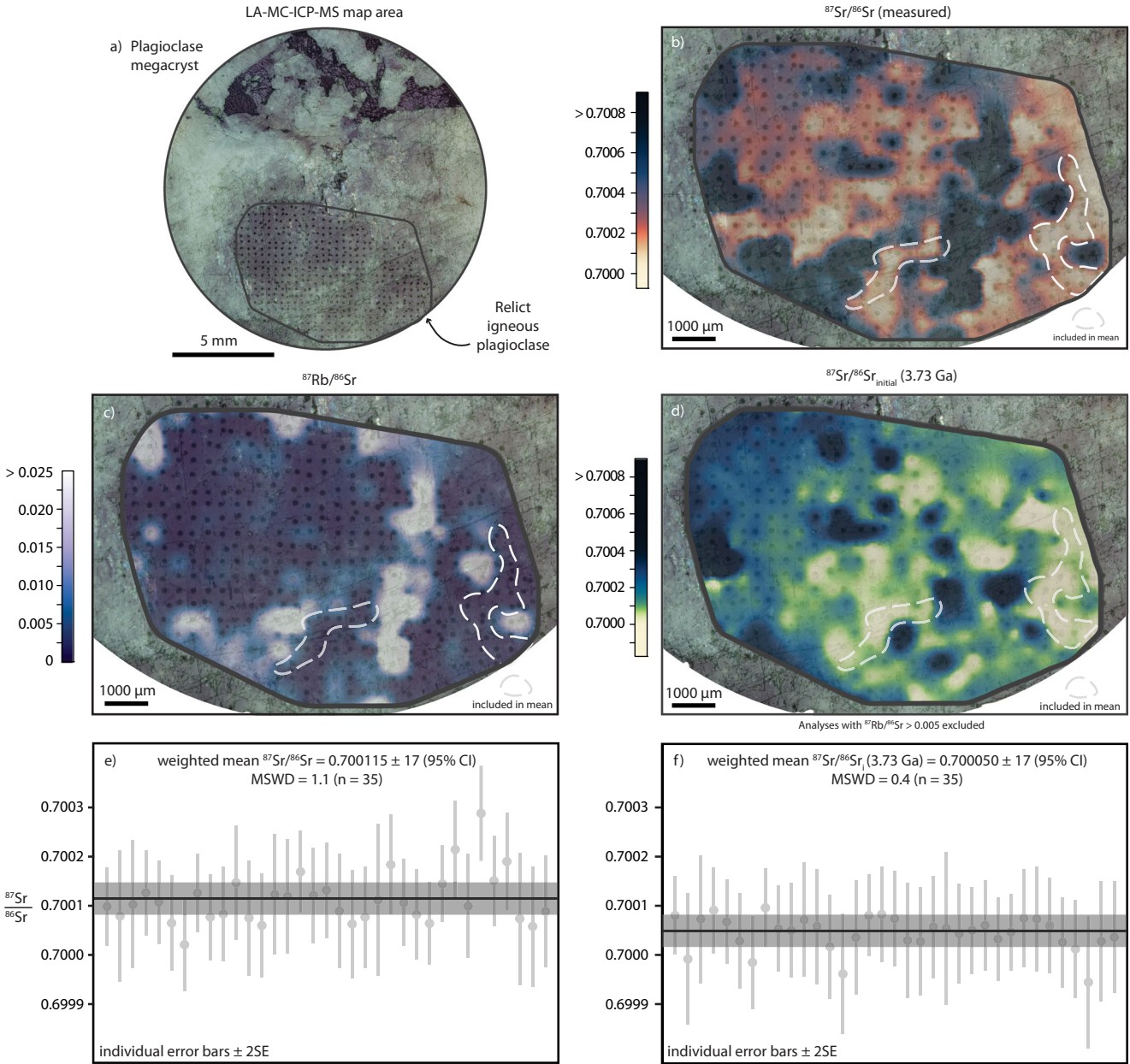

**Fig. 2 | Strontium isotope mapping of Manfred Complex plagioclase. a** Optical image of plagioclase megacryst (An$_{80}$) from the 3.73 Ga Manfred Complex, Narryer Terrane, Western Australia, with the area analysed by LA-MC-ICPMS enclosed by the grey line (laser ablation spots visible within this area). Fractures and altered areas appear optically white in colour relative to the fresher domains, which appear dark grey. **b** Interpolated map of measured $^{87}Sr/^{86}Sr$. **c** Interpolated map of

$^{87}Rb/^{86}Sr$. **d** Interpolated map of $^{87}Sr/^{86}Sr_{initial}$ (at 3.73 Ga). Analyses with elevated (>0.005) $^{87}Rb/^{86}Sr$ are excluded from this map to avoid overcorrecting the data. **e** Weighted mean of measured $^{87}Sr/^{86}Sr$ calculated for the outlined map domains in (**b**). Domains were selected on the basis of clusters of adjacent low $^{87}Sr/^{86}Sr$ and $^{87}Rb/^{86}Sr$ measurements. **f** Weighted mean of $^{87}Sr/^{86}Sr_{initial}$ (at 3.73 Ga) calculated for the outlined map domains in (**d**).

concentrations in the unknowns (~100–200 µg/g), a Jet-X sample-skimmer cone configuration was employed during this study, with ~10 mL/min N$_2$ added to improve sensitivity and a total of 1 L/min He introduced to the cup and cell as the carrier gas. Two-pulse pre-ablations were applied to all samples and standards prior to analysis. Typical analytical conditions included 110 µm circular spots for unknowns, 50–110 µm circular spots for reference materials, 4–5 J/cm$^2$ fluence, and a repetition rate of 10–15 Hz. Every 60 s of ablation time was followed by a 70 s washout period, the final 30 s of which was used to determine the gas blank. Unknowns were bracketed with standards throughout. An in-house plagioclase standard, Australian Andesine, was used as the primary reference material. Secondary standards included the plagioclases AMNH-107160, G29958, Hrappsey, BDL, and

Lake County (characterised in-house), and ATHO-G glass was used to calibrate Rb/Sr fractionation[66–68]. Data were reduced using Iolite software and the strontium isotope DRS of Mulder et al.[66], with a correction for Rb/Sr fractionation applied[66,69]. Results of reference material analyses, including TIMS analysis of in-house standards, are given in Supplementary Data 3–4.

Time integrated signals were monitored for anomalous spikes in $^{87}Rb/^{86}Sr$, $^{88}Sr$ and $^{87}Sr/^{86}Sr$, indicating the intersection of inclusions or alteration zones, and were cropped where necessary to avoid mixed analyses (e.g. Supplementary Fig. 6). Analyses with anomalously high $^{87}Rb/^{86}Sr$ (>3 SD from the mean) were excluded, and the weighted mean $^{87}Sr/^{86}Sr$ and $^{87}Sr/^{86}Sr_i$ were determined for each sample using the published U-Pb ages for each complex[32–34] (Supplementary

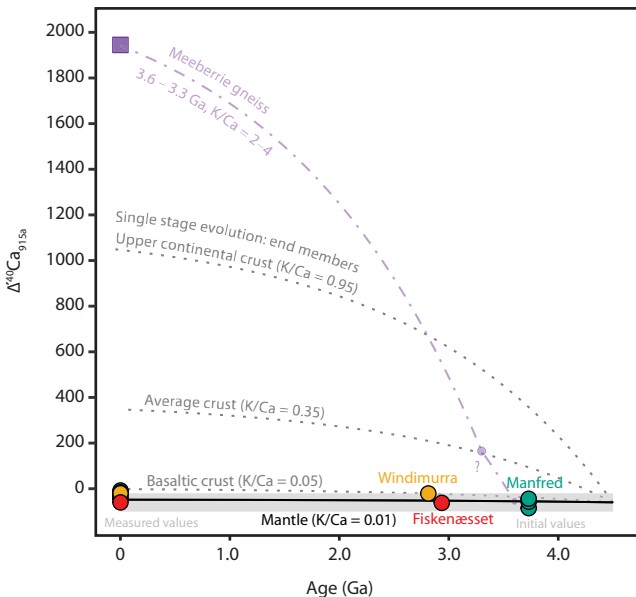

**Fig. 3 | Calcium isotope results.** Ca isotope evolution curves of the mantle and single-stage crustal end-member models[24]. Uncertainties are plotted as ±2SE and are smaller than the symbols. Calcium isotopes are particularly sensitive to the presence of ancient felsic crust; a 3.6–3.3 Ga sample of the Meeberrie gneiss from Western Australia (K/Ca of 2–4) has a modern $\Delta'^{40}Ca_{915a}$ of 1951 ± 14[25,85,86], and is plotted for comparison to represent the highly radiogenic composition of potential ancient felsic crustal contaminants. The Meeberrie gneiss has multiple age components and appears to have had a multi-stage Ca isotopic evolution[25]; the question mark indicates that the plotted curve, with an increase in K/Ca from ~2 to ~4 at 3.3 Ga, represents only an estimate of its evolution. The Windimurra, Fiskenæsset, and Manfred Complex plagioclase samples plot within the uncertainty of the mantle value, ruling out significant felsic crustal contamination. Full results are available in Supplementary Data 5.

Figs. 7–9). For samples where the MSWD ≈ 1, the weighted means of all analyses were used, calculated using the method of Vermeesch et al.[70]. For samples where MSWD » 1, the weighted mean of a population of spatially related and least radiogenic analyses with MSWD ≈ 1 was used, where possible. If the data were too scattered and such a population could not be identified, this was taken to indicate the measured $^{87}Sr/^{86}Sr$ ratio was unlikely to represent the primary magmatic isotopic composition due to isotopic disturbance, and such data were excluded.

Analytical conditions for strontium isotope maps were the same as for point analyses. Plagioclase strontium isotope maps of one Manfred Complex megacryst were constructed using a grid of 110 μm circular point analyses spaced approximately every 300 μm, interpolated using a standard kriging method in the Surfer software package and coloured using the palettes of Crameri (2023)[71]. To construct the age-corrected maps, analyses with anomalously elevated Rb ($^{87}Rb/^{86}Sr > 0.005$) were excluded to avoid large overcorrections due to alteration.

### Micro-mill TIMS

Samples were prepared at the University of Western Australia in the form of polished epoxy-mounted rock pieces and polished blocks (from which the polished thick sections previously analysed by LA-MC-ICPMS were cut). Plagioclase domains that were calcic, optically clear, and recorded unradiogenic $^{87}Sr/^{86}Sr$ and low $^{87}Rb/^{86}Sr$ ratios by LA-MC-ICPMS were targeted for micro-milling and solution isotope analysis. Optical, BSE, and CL images of plagioclase samples were evaluated so as to avoid fractures or alteration at the surface.

Micro-milling was performed using a New Wave MicroMill at the University of Bristol, equipped with a 500 μm diameter tungsten carbide drill bit. The sample and drill bit were thoroughly cleaned

three times in an ultrasonic bath in 18 MΩ·cm ultra pure water (UPW), once in high purity ethanol, and dried with a lint-free kimwipe prior to milling for each aliquot. A droplet of UPW was placed on the sample surface prior to milling, creating a slurry that was then collected via pipette into a clean 7 mL perfluoroalkoxy (PFA) beaker. Due to low Sr concentrations (~100–200 μg/g), four individual holes 400 × 400 μm in size were drilled per aliquot, clustered around an LA-MC-ICPMS Sr spot location, to achieve the optimal minimum loads of Sr (>25 ng) and Ca (>25 μg). To minimise the inclusion of any pervasive fine alteration in Manfred Complex sample 12N8, only one individual hole 400 × 400 μm in size was drilled per aliquot corresponding to the least radiogenic domains on the LA-MC-ICPMS Sr map, resulting in reduced loads of ~12 ng Sr and ~8 μg Ca (Supplementary Fig. 5). Aliquots of Telica plagioclase[72] and of a modern marine shell were utilised as secondary reference materials following the same method outlined above. Procedural blanks were collected by lowering and engaging the drill bit into a droplet of UPW above the surface of a cleaned sample and pipetting the UPW in the same fashion as an unknown sample.

Two plagioclase crystals, 'Australian andesine' and 'Lake County' plagioclase, were characterised for use as in-house LA-MC-ICPMS strontium isotope standards. 'Australian andesine' is a gem-quality plagioclase crystal (An47-49) approximately 1 cm in length of unknown provenance sourced from a gem dealer, with a strontium concentration of 1048 ± 17 μg/g (2 SD, n = 6). "Lake County" is a gem-quality 'sunstone' plagioclase crystal (An66-68) from a porphyritic basalt flow in Lake County, Oregon, with a strontium concentration of 588 ± 11 μg/g (2 SD, n = 18). Three aliquots of randomly selected crystal fragments weighing between 0.5 and 1 mg were picked for each sample (corresponding to loads of approximately 500 ng Sr per aliquot) and were ultrasonicated in purified acetone and UPW prior to dissolution.

All samples were dissolved in 1000 μL of super pure grade concentrated $HNO_3$ and HF (1:1), dried down, and redissolved in 550 μL of 3 M $HNO_3$. A 50 μL aliquot of each sample was reserved for trace element analysis. The strontium fraction was then separated using Sr Spec resin[73]. The 'waste' fraction from the Sr column was put aside for Ca purification. Strontium fractions were dried and loaded onto single standard purity Re filaments with $TaCl_5$ used as an activator to improve ionisation efficiency, following the modified method of Birck et al.[74].

Strontium isotope ratios were measured in a multidynamic collection mode using a Thermo-Finnigan Triton TIMS instrument at the University of Bristol[75,76]. Amplifier gains were calibrated before each session, and each filament was heated manually and analysed for up to 200 cycles with 4.194 s integration time per cycle. Strontium 87 beams were corrected for isobaric $^{87}Rb$ by monitoring $^{85}Rb$ and using a $^{85}Rb/^{87}Rb$ value of 2.59265[77]. Samples were corrected for mass fractionation by internal normalisation to $^{86}Sr/^{88}Sr = 0.1194$[78], using the exponential mass fractionation law[79]. Reference material NIST SRM 987 was measured, with both pure and column-processed aliquots analysed in every session. Procedural blanks were in the range of 60–100 pg. All data were normalised to the NIST SRM 987 $^{87}Sr/^{86}Sr$ value of 0.710248[80].

### Calcium isotopes

Post strontium chemistry, selected samples with unradiogenic $^{87}Sr/^{86}Sr$ were processed through calcium chemistry at the University of Bristol using the procedure outlined by Lewis et al.[25]. Radiogenic calcium isotope ratios were measured on the Proteus CC-MC-ICPMS/MS instrument at the University of Bristol in low mass resolution, following a modified version of the procedure of Lewis et al.[25]. Analyses were bracketed with measurements of NIST SRM 915a solution, with total Ca signal intensities of unknowns matched to within ±5% of the bracketing solution. Analyses of basaltic powders BHVO and GSP-2 and a modern

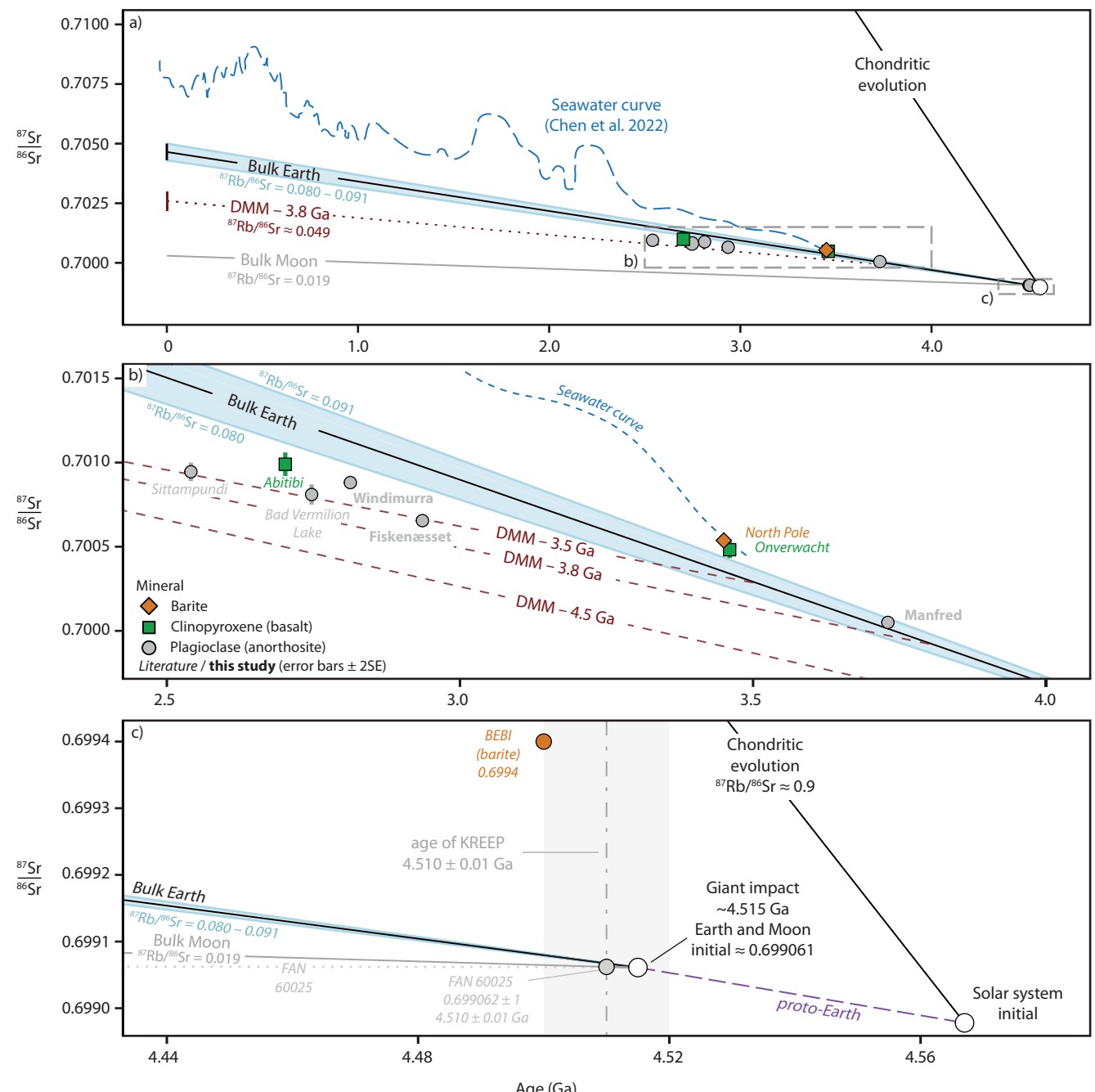

**Fig. 4 | Strontium isotope evolution of the early Earth. a** Evolution of the estimated modern bulk Earth projected back to the formation of the Earth and Moon, with the depleted mantle[19] and seawater curves[59] plotted for comparison. The depleted mantle curve is projected from a depleted modern mantle (DMM) value of $0.70263 \pm 44$[19] to the bulk Earth at 3.8 Ga. The bulk Earth is Rb/Sr depleted compared to chondritic evolution, and the bulk Moon is significantly more Rb/Sr depleted[44]. **b** Archaean strontium mantle evolution. Published high-precision Sr isotope analyses of minerals from Archaean mafic rocks that have both unradiogenic measured and initial $^{87}Sr/^{86}Sr$ are included for comparison[22,87–91] (Supplementary Data 6). Strontium isotope studies of inclusions in Archaean minerals have not been included due to the lower precision and large radiogenic ingrowth corrections applied[92,93]. Data from this study and the literature indicate that the depleted mantle signature became established relatively late in Earth history for

strontium isotopes. The North Pole barite and Onverwacht basalt, samples previously used to define the bulk Earth Sr isotope composition, appear to reflect seawater compositions rather than pristine mantle. The Manfred Complex plots within the range of estimated bulk Earth compositions, providing an upper limit for the composition of the primitive mantle of 0.700050 at 3.73 Ga. **c** Model for the initial $^{87}Sr/^{86}Sr$ ratio of the Earth and Moon, assuming Rb-Sr equilibration at the time of the giant impact, taken here to occur at ~4.515 Ga[52–54]. The retarded Sr isotope evolution of the bulk Moon is a result of volatile loss (Rb) at the time of impact. A shared initial ratio of $0.699061 \pm 1$ at 4.515 Ga is consistent with the best preserved early terrestrial and lunar data as measured in ferroan and Archaean anorthosites[15]. The earlier Bulk Earth Best Initial (BEBI) value of 0.6994 determined from barite[22] is plotted for comparison.

marine shell were included as secondary reference materials. Each measurement was internally normalised to a $^{42}Ca/^{44}Ca$ ratio of 0.31221 and externally normalised to the mean of the two bracketing SRM 915a measurements. Each analysis consists of the mean of 8–16 bracketed

measurements of a single sample solution, except for sample 12N8, where 4–8 bracketed measurements were performed per aliquot due to the smaller sample volumes collected. Procedural blanks were in the range of <1–2% total blank contribution.

Two aliquots, 21MBY04 and 151033, recorded anomalously elevated counts of $^{87}Sr^{++}$, indicating that these samples were contaminated with excess Sr either during the Ca column chemistry or dilution process[25]. Excess K was also detected in these two aliquots and $\Delta'^{43}Ca_{915a}$ measurements were elevated (Supplementary Fig. 10). Due to contamination, these two samples were therefore excluded.

## Radiogenic ingrowth corrections

Prior to Sr separation via Sr-specific extraction chromatography, a 50 μL aliquot of each sample, dissolved in 3 M $HNO_3$, was reserved for elemental and isotopic analysis. To prepare a working stock solution, 950 μL of 2% $HNO_3$ (v/v) was added to the aliquot, yielding a 1 mL solution. A 1:100 dilution of this stock was prepared for preliminary quantification of calcium (Ca) concentrations to guide final dilution volumes. The remaining 1 mL stock solutions were evaporated to dryness in acid-cleaned 7 mL Savillex® PFA vials on a hotplate at 120 °C. Dried residues were redissolved in 2% $HNO_3$ to achieve an approximate final Ca concentration of 1 μg/mL.

Elemental and isotopic measurements were performed at the Centre for Microscopy, Characterisation and Analysis, University of Western Australia using a Thermo Fisher Scientific Element XR sector field inductively coupled plasma mass spectrometer (SF-ICP-MS), equipped with a Peltier-cooled cyclonic spray chamber and a 200 μL/min PFA nebuliser. A JET nickel sampler cone and a nickel X-cone were used throughout the analysis to maximise ion transmission and sensitivity. Instrument slits were configured to low resolution ($R \sim 300$) for rubidium (Rb) and strontium (Sr), and medium resolution ($R \sim 4000$) for potassium (K) and calcium (Ca).

Blanks were analysed every 10 unknown samples. Blank subtraction was performed using the average signal of the preceding and subsequent blanks for each isotope. Final elemental concentrations of K, Ca, Rb, and Sr were calculated by external calibration using serial dilutions of single-element standards. Calibration curves were generated independently for each element from 4 to 5 concentration points spanning the expected range of unknowns. All calibration curves exhibited linearity with $R^2 > 0.999$.

Instrumental operating parameters for the Element XR were optimised to maximise ion sensitivity. Supplementary Table 2 lists the instrument parameters used for data acquisition. Limits of detection (LOD) for each element were calculated using the standard deviation of replicate blank measurements and the slope of the corresponding calibration curve:

$$LOD = (3.3 \times \sigma_{blank})/m$$

Where $\sigma_{blank}$ is the standard deviation of the blank signal (cps), and $m$ is the slope of the calibration curve (cps/ppt). The LOD values are reported in parts per trillion (ppt) in Supplementary Table 3. For samples with K and or Rb concentrations below the LOD, maximum values for $^{87}Rb/^{86}Sr$ and $^{40}K/^{44}Ca$ calculated using the LOD for Rb or K are reported.

Isotopic ratios were determined using determined elemental concentrations and calculated assuming IUPAC-recommended relative atomic masses and natural isotopic abundances[81,82]. These values are summarised in Supplementary Table 4. Final isotope ratio relative uncertainties were calculated through quadrature propagation of the measured isotope RSDs.

Measured $^{87}Rb/^{86}Sr$ and $^{40}K/^{44}Ca$ ratios were used to determine the initial $^{87}Sr/^{86}Sr$ and $\Delta^{40}Ca_{915a}$ ratios for the micro-milled samples (Figs. 3–4, Table 1, Supplementary Data 4–5, Supplementary Fig. 11). Initial $^{87}Sr/^{86}Sr$ were determined using the decay constants $\lambda(^{87}Rb) = 1.3972 \times 10^{-11}$ $a^{-1}$, and initial $\Delta^{40}Ca_{915a}$ were determined using the decay constant $\lambda(^{40}K) = 5.51 \times 10^{-10}$ $a^{-1}$ and branching ratio of 0.8952[10,83].

## Data availability
All data supporting the findings of this study are presented within the paper and in the Supplementary Information and Data files. These data are available at https://doi.org/10.6084/m9.figshare.28794356.

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

## Acknowledgements

We acknowledge the Wajarri Yamaji and Badimia people as the Traditional Owners of the lands on which part of this research took place. We thank the late J. Myers for providing the Greenland samples for this study, and E. Koojiman, J. Mulder, J. Hammerli, and J. Hanchar for providing isotopic standards. M. Boyce thanks M. Roberts, S. Barnes, L. Schoneveld, C. Coath, and the ANSTO staff for assistance with microanalysis, and acknowledges financial support from the Robert and Maude Gledden Postgraduate Scholarship. Field logistical support was provided by the Geological Survey of Western Australia. We acknowledge the facilities, scientific, and technical assistance of the Australian Microscopy & Microanalysis Research Facility at the Centre for Microscopy, Characterisation & Analysis, The University of Western Australia, a facility funded by the University, State and Commonwealth Governments. Imaging was undertaken on the XFM beamline (proposal 19325) at the Australian Synchrotron, part of ANSTO. T.I. publishes with permission from the Executive Director of the Geological Survey of Western Australia. This work was supported by Australian Research Council grant DP200103208 (T.K., T.E., S.W.). Strontium isotope analyses at UWA were conducted with instrumentation funded by the Australian Research Council (LE100100203 and LE150100013). We thank L. Ashwal and J. Kendrick for thoughtful comments and suggestions.

## Author contributions

T.K., T.E., S.W. and M.B. devised the study. T.K., M.B. and T.I. were involved with sample collection. M.B., C.F., D.B., A.S. and J.L. contributed to sample characterisation and chemical and isotopic analysis. M.B. evaluated the data, synthesised the results, and prepared the figures and first draft of this manuscript, with input from all coauthors. All authors contributed to the editing of the manuscript.

## Competing interests

The authors declare no competing interests.
