## [Transparent Peer Review file · Nature Communications]

Coupled strontium-calcium isotopes in Archean anorthosites reveal a late start for mantle depletion

Corresponding Author: Ms Matilda Boyce

Version 0:

Reviewer comments:

Reviewer #1

(Remarks to the Author)

The paper constrains the Sr isotopic composition of the Earth's very early mantle, with implications for the equilibration of the Earth-Moon system at the time of the giant impact event. By utilizing high-precision isotopic measurements in Archean anorthosites as old as 3.75 Ga, the authors have detected the lowest measured initial Sr ratio for terrestrial plagioclase. The paper also finds that the onset of mantle depletion, presumably caused by significant crust formation, occurred only after about 3.5 Ga. These are important findings of global and planetary significance, and I would very much like to see this well-written paper published. Below I offer some aspects for the authors to consider.

The paper really has two themes- the onset of terrestrial mantle depletion and Earth-Moon equilibration during the giant impact event. But the title only mentions the first of these. Maybe consider broadening the title to include both themes? Both are of global importance.

I wonder if it would be good to discuss the type of terrestrial mantle being constrained here. Excluding sub-continental mantle (because the Archean anorthosites studied here likely formed in oceanic environments), the present Earth has sub-MOR mantle, oceanic arc mantle wedge, deep plume mantle, and others. This relates to the tectonic setting of Archean anorthosites (ridge? plume? subduction?) and their occurrence in Archean greenstone belts. Supplying some specifics to this would be useful to those studying temporal evolution of specific mantle types.

Were concentrations of Rb and Sr measured? I can't find these in the tables. These would be useful for the calculation of initial ratios. Or was $^{87}\text{Rb}/^{86}\text{Sr}$ measured directly? If so, how was overlap at mass 87 handled?

Secondary epidote is mentioned as responsible for radiogenic Sr seen in some plagioclase analyses. But the terrific images of plagioclase grains don't show any epidotes. I can't see the epidote in Fig. 1 that was advertised at lines 116-117. How was the presence or absence of epidote constrained (discussed at 126-127)?

Lewis Ashwal
Johannesburg
28 May 2025

Reviewer #2

(Remarks to the Author)

"Coupled strontium-calcium isotopes in Archean anorthosites reveal a late start for mantle depletion" by Boyce et al.

The authors present new Sr isotope data from plagioclase in Archean anorthosite and leucogabbro, with the goal of better constraining the Sr isotope evolution of early Earth. The knowledge gap is clearly defined – Sr isotope data from Archean samples are scarce relative to other radiogenic systems, and this information is needed to establish the Sr isotope evolution of depleted mantle and reconcile the existing Sr records of the Earth and the Moon. Given the high susceptibility of Sr isotopes to be modified by alteration, the authors took several steps to identify the most pristine areas for in situ analysis, including petrography, monitoring Ca isotopes, LA-ICP-MS mapping, and μXRF mapping by Synchrotron. Boyce et al. use

their Sr isotope data to calculate a new estimate for Earth's initial $^{87}\text{Sr}/^{86}\text{Sr}$, which shows agreement with the Sr isotope record of the Moon, and they further conclude that their data support a later start for Earth's depleted mantle curve (ca. 3.8 Ga).

The importance of the research is clear, the conclusions are supported by the data, and the paper is very well-written. I do not have the expertise to evaluate the soundness of the analytical protocols employed for Sr and Ca isotope analysis, but it appears that the authors followed best practices and described the methods clearly. A big challenge in this work was avoiding areas where the Rb–Sr system was modified post-crystallization, and I believe that the authors have taken the appropriate steps to address this—I appreciate the many images included showing the plagioclase crystals and analysis locations. Overall, I recommend publication of this manuscript after minor revisions.

General comment:

Although I believe that the authors have avoided alteration in their samples and the Ca isotopes were useful for identifying contamination by felsic crust, the authors should comment on the possibility that the Sr isotope composition of the mantle source could have been modified by subduction-related fluids. Some researchers have suggested that Archean anorthosites primarily formed in subduction zone settings (e.g., Sotiriou et al., 2024, Earth-Science Reviews). Would the Ca isotope analyses also help to monitor for contamination by altered oceanic crust and related fluids at the source?

Line-by-line comments:

L373: Lake Country or Lake County?

L414: It looks like this isn't displaying properly here and elsewhere, should be HNO with 3 as the subscript, right?

L481: Double-check references, this one is formatted differently from the others.

Figures:

The figure captions for Figs 1 and 3 mention uncertainties $\pm 2\text{SE}$, however these are not apparent on the figure. Are the error bars smaller than the symbols? If so, it would be helpful to mention this.

For Fig 4d, there are some very light grey error bars that are difficult to see. The authors should darken these to make them more visible.

Supplementary material:

Supplementary Figure 7 – the caption mentions outliers, however there do not appear to be any.

Supplementary Figure 9 – “Samples have MSWD $\gg 1$, indicating all are disturbed to some degree” What about the three with MSWD < 2 ?

Jillian Kendrick
Saint Mary's University
Halifax, Canada

Version 1:

Reviewer comments:

Reviewer #1

(Remarks to the Author)

I have read through the author responses to the reviewer comments- both mine and those of reviewer #2.

I am satisfied that the authors have addressed our concerns adequately.

REVIEWER COMMENTS

Replies to reviewer comments: red text. Matilda Boyce (September 2025)

Reviewer #1 (Remarks to the Author):

The paper constrains the Sr isotopic composition of the Earth's very early mantle, with implications for the equilibration of the Earth-Moon system at the time of the giant impact event. By utilizing high-precision isotopic measurements in Archean anorthosites as old as 3.75 Ga, the authors have detected the lowest measured initial Sr ratio for terrestrial plagioclase. The paper also finds that the onset of mantle depletion, presumably caused by significant crust formation, occurred only after about 3.5 Ga. These are important findings of global and planetary significance, and I would very much like to see this well-written paper published. Below I offer some aspects for the authors to consider; I have also made some comments and queries on an annotated version of the manuscript.

Thank you, we are most grateful for the encouraging comments.

The paper really has two themes- the onset of terrestrial mantle depletion and Earth-Moon equilibration during the giant impact event. But the title only mentions the first of these. Maybe consider broadening the title to include both themes? Both are of global importance.

Thank you, we have discussed this suggestion, and the two themes are linked. However, we have decided to keep the title as is, to focus on the mantle evolution aspect, which we think is more novel given many recent studies documenting isotopic equilibration between Earth and Moon.

I wonder if it would be good to discuss the type of terrestrial mantle being constrained here. Excluding sub-continental mantle (because the Archean anorthosites studied here likely formed in oceanic environments), the present Earth has sub-MOR mantle, oceanic arc mantle wedge, deep plume mantle, and others. This relates to the tectonic setting of Archean anorthosites (ridge? plume? subduction?) and their occurrence in Archean greenstone belts. Supplying some specifics to this would be useful to those studying temporal evolution of specific mantle types.

Yes, this is a good point and is something we have put much thought to. However, there are a couple of difficulties here: firstly, these different mantle types are likely to be isotopically less well resolved from each other in the Archean, having had less time to evolve. Secondly, these modern mantle types are linked to geodynamic processes which may or may not have been operating during the Archean; this a topic of much long-standing discussion but is not something we aim to address with this paper. The final challenge is that there is still no consensus on the specific geodynamic setting in which Archean anorthosites formed beyond an oceanic environment. For example, there has been much debate over whether the Fiskensset Complex formed from a hydrous or anhydrous magma, in an arc-like or non-arc setting (recent examples being Polat et al. 2014, Linnebjerg et al. 2025). The lack of a direct modern analogue for Archean anorthosites could also suggest they

formed in tectonic environments unlike any modern setting. For this reason, we have been cautious with interpreting the data as being representative of specific mantle types, instead stating that the lowest determined $^{87}\text{Sr}/^{86}\text{Sr}$ represent a maximum (most radiogenic) estimate of the depleted mantle at the time. This also takes into consideration how the timing and extent of mantle depletion is unlikely to have been uniform globally.

We hope that with time, a larger dataset of robust strontium isotope measurements become available for Archean mantle-derived rocks, akin to the quality-filtered compilations used to construct recent Hf and Nd depleted mantle curves (e.g. Vervoort and Kemp 2024). The sensitivity of Sr to fluid-driven processes has the potential to possibly provide unique information to the Hf and Nd mantle curves and may reveal new insights on early mantle heterogeneity that can more clearly be linked to tectonic settings. This could also help to address the question of the tectonic environment of Archean anorthosites.

We have expanded the sentence on line 82 to briefly state this in the main text.

Were concentrations of Rb and Sr measured? I can't find these in the tables. These would be useful for the calculation of initial ratios. Or was $^{87}\text{Rb}/^{86}\text{Sr}$ measured directly? If so, how was overlap at mass 87 handled?

Yes, concentrations of Rb and Sr were measured from the micro-milled aliquots (sub-sampled prior to column chemistry) and these were used to calculate the TIMS initial ratios. Mass 85 was measured to determine Rb concentrations, and mass 88 measured for Sr concentrations. We have added the Rb and Sr concentrations in ppm to the solution results in the Supplementary File (Table 4. Sr TIMS). For the LA-MC-ICPMS measurements, $^{87}\text{Rb}/^{86}\text{Sr}$ were measured directly, calculated using the Lolite Universal Sr Isotope data reduction scheme of Mulder et al. (2023). This method handles overlap on mass 87 by measuring Rb on mass 85 and then determines ^{87}Rb by multiplying ^{85}Rb by the natural $^{87}\text{Rb}/^{85}\text{Rb}$ isotopic ratio. Initial ratios were then calculated for all analyses.

Secondary epidote is mentioned as responsible for radiogenic Sr seen in some plagioclase analyses. But the terrific images of plagioclase grains don't show any epidotes. I can't see the epidote in Fig. 1 that was advertised at lines 116-117. How was the presence or absence of epidote constrained (discussed at 126-127)?

Although no images of this were shown, most Manfred Complex leucogabbros and anorthosites are extensively altered, containing abundant coarse-grained secondary epidote within relict plagioclase (also described by Myers 1988a). The plagioclase presented in Fig. 1 is an example of a better-preserved Manfred Complex sample where this alteration is less progressed, and indeed, epidote is difficult to identify at the scale of this figure. In the XFM maps, optically "dusty" altered areas of plagioclase (e.g. now labelled on the right side of Fig. 1c) contain fine grains of a calcium-rich secondary mineral interpreted as epidote based on its composition and the presence of epidote in other samples. However, it would be more accurate to state that these altered domains likely contain an assemblage of secondary minerals, and the phrasing has been adjusted to reflect this.

Lewis Ashwal
Johannesburg
28 May 2025

Annotations (PDF file):

Line 15: **Corrected.**

Line 17: **Done.**

Line 21: **Done.**

Line 24: **Done.**

Line 25: **Added.**

Line 26: **Added.**

Line 40: **This is a good point. Baddeleyite wasn't mentioned here as currently there's very little isotopic data available for it from the Archean. Although baddeleyite is an excellent candidate for dating and for Lu-Hf analysis, the focus so far has been on zircon mainly because baddeleyite is challenging to recover due to its low abundance, morphology and small grain size. Some Archean anorthosite samples, including from the Manfred Complex (Kinny et al. 1988), are sufficiently fractionated to contain small quantities of zircon, which is comparatively simpler to recover. Baddeleyite Lu-Hf in silica-undersaturated Archean mafic rock is certainly an avenue worth exploring further in the future, particularly if an improved method of separating it is devised, or analysis is performed in-situ (e.g. via SIMS).**

Line 52: **Added a brief explanation of the Rb/Sr estimates.**

Line 54: **Done.**

Line 73: **Yes, both present-day and ancient mantle estimates are included within this range of values – I have edited this sentence for clarity. The typical precision achievable for Ca isotope measurement is currently greater than the total amount of estimated terrestrial mantle evolution, resulting in a near-constant mantle value through time.**

Line 73 (2): **Edited to include the range of estimated values.**

Line 78: **Yes, Ca isotopes are used here primarily to filter for mantle-like signatures. This is useful because, for example, if a sample returned an unradiogenic Sr signature but a radiogenic Ca signature, this could indicate the presence an early highly depleted mantle source mixed with an ancient crustal component. This would suggest the unradiogenic Sr isotope signature is not representative of the isotopic composition of the mantle source. Since the Ca isotopes returned mantle-like values, we can rule out such a scenario.**

Line 90: Yes, this is what was meant here. Some additional explanation has been added to the text.

Line 107: All data were age corrected using the measured $^{87}\text{Rb}/^{86}\text{Sr}$, but these corrections were negligible in the well-preserved plagioclase as the measured $^{87}\text{Rb}/^{86}\text{Sr}$ were very low. This is to avoid the effects illustrated in Fig. 1, where addition of secondary ^{87}Rb leads to unrealistically low age-corrected $^{87}\text{Sr}/^{86}\text{Sr}$. This section has been reworded for clarity.

Line 117: A label now added to Fig 1 to indicate the altered areas containing secondary mineral assemblages, which are fine-grained and most apparent in the XFM images. Changed the text from epidote to secondary mineral assemblages.

Line 127: As above: labelled the interpreted secondary alteration.

Line 180: Done.

Line 256: Thank you for pointing this out; the figure caption has been corrected. The inflection point in the Meeberrie Gneiss curve represents the two age populations in that sample of the gneiss: the younger ~3.3 Ga component has a higher K/Ca of ~4 than the older ~3.6 Ga component (K/Ca of ~2) (Myers 1988b). The question mark is to indicate that the plotted curve is only an estimate of how the Meeberrie Gneiss may have evolved to its modern $\Delta'_{40}\text{Ca}_{915a}$ isotope composition. This is because the Meeberrie Gneiss appears to have had a multi-stage Ca isotope evolution, as the modern $\Delta'_{40}\text{Ca}_{915a}$ value is slightly lower than the value returned when a calculation is made for radiogenic ingrowth using the age of the rock and the bulk composition, as discussed by Lewis et al. (2022) who performed the measurement on this sample (14WA24). The Meeberrie Gneiss is plotted as a comparison to the unradiogenic measured samples to show that Archean felsic crust can contain very radiogenic modern-day $\Delta'_{40}\text{Ca}_{915a}$, and how it may have evolved does not affect the results.

Reviewer #2 (Remarks to the Author):

“Coupled strontium-calcium isotopes in Archean anorthosites reveal a late start for mantle depletion” by Boyce et al.

The authors present new Sr isotope data from plagioclase in Archean anorthosite and leucogabbro, with the goal of better constraining the Sr isotope evolution of early Earth. The knowledge gap is clearly defined – Sr isotope data from Archean samples are scarce relative to other radiogenic systems, and this information is needed to establish the Sr isotope evolution of depleted mantle and reconcile the existing Sr records of the Earth and the Moon. Given the high susceptibility of Sr isotopes to be modified by alteration, the authors took several steps to identify the most pristine areas for in situ analysis, including petrography, monitoring Ca isotopes, LA-ICP-MS mapping, and μXRF mapping by Synchrotron. Boyce et al. use their Sr isotope data to calculate a new estimate for Earth’s initial $^{87}\text{Sr}/^{86}\text{Sr}$, which shows agreement with the Sr isotope record of the Moon, and they further conclude that their data support a later start for Earth’s depleted mantle curve (ca. 3.8 Ga).

The importance of the research is clear, the conclusions are supported by the data,

and the paper is very well-written. I do not have the expertise to evaluate the soundness of the analytical protocols employed for Sr and Ca isotope analysis, but it appears that the authors followed best practices and described the methods clearly. A big challenge in this work was avoiding areas where the Rb–Sr system was modified post-crystallization, and I believe that the authors have taken the appropriate steps to address this—I appreciate the many images included showing the plagioclase crystals and analysis locations. Overall, I recommend publication of this manuscript after minor revisions.

Thank you, we are most grateful for these comments.

General comment:

Although I believe that the authors have avoided alteration in their samples and the Ca isotopes were useful for identifying contamination by felsic crust, the authors should comment on the possibility that the Sr isotope composition of the mantle source could have been modified by subduction-related fluids. Some researchers have suggested that Archean anorthosites primarily formed in subduction zone settings (e.g., Sotiriou et al., 2024, Earth-Science Reviews). Would the Ca isotope analyses also help to monitor for contamination by altered oceanic crust and related fluids at the source?

This is a good suggestion. Beyond a mantle-derived and oceanic origin, many intriguing questions remain as to the geodynamic setting in which Archean anorthosites formed. Some authors indeed suggest an island arc or back-arc basin setting for Archean anorthosite complexes, including for two of the intrusions studied here, the Fiskensæset Complex (e.g. Polat et al. 2009; Polat and Longstaffe 2014) and Windimurra Igneous Complex (e.g. Lowrey et al. 2024). However, this is not the consensus, with other researchers suggesting alternative geodynamic settings including oceanic plateaus, rifts, stagnant-lid or plume-type origins (e.g. Ivanic et al. 2015; Rowe and Kemp 2020; Linnebjerg et al. 2025). This is also tied to the ongoing discussion on when modern plate tectonic processes commenced.

Modern island arc $^{87}\text{Sr}/^{86}\text{Sr}$ is thought to predominantly reflect mixing of depleted mantle and seawater Sr sourced from subducted altered oceanic crust and sediment (e.g. Bednarick et al. 2024). However, this radiogenic signature (0.7035–0.7045) represents only a modest perturbation from depleted mid-ocean ridge basalts (0.7025) and is indistinguishable from many intra-plate oceanic basalts (e.g. so-called Focal Zone basalts 0.703–0.7035). If subduction processes were operating during the Archean, the elevated $^{87}\text{Sr}/^{86}\text{Sr}$ signatures of island arcs would be even more subtle compared to those measured in modern rocks because of the shorter time for reservoirs to develop isotopic differences. Modern seawater is much more radiogenic for $^{87}\text{Sr}/^{86}\text{Sr}$ (~0.7092) than the modern depleted mantle (~0.7025), but at ~2.8 Ga, this difference was much smaller, with ~0.7020 in seawater (Chen et al. 2022) vs ~0.7006 in the mantle by our estimates, and by ~3.7 Ga, we estimate seawater likely had a near-identical isotopic composition to the mantle. Any potential mixing between depleted mantle and seawater Sr in the source of Archean anorthosites would result in isotopic enrichment, but at a lesser extent compared to modern subduction rocks.

We believe radiogenic Ca isotopes are unlikely to provide a good monitor of subduction processes. Compared to strontium, calcium is not very incompatible and is unlikely to be efficiently transported by fluids, and the isotopic differences between reservoirs are much smaller for the K-Ca system. Modern seawater has an estimated $\Delta^{40}\text{Ca}_{915a}$ of ca. 0 (Caro et al. 2010; Lewis et al. 2022), vs ca. -60 in the mantle (Wang et al. 2024); these reservoirs are often unresolvable with the typical precision currently achievable for radiogenic Ca measurements. Archean seawater was likely even closer or identical in composition to the mantle for $\Delta^{40}\text{Ca}_{915a}$. Radiogenic Ca isotopes are therefore unlikely to detect mixing between depleted mantle and altered oceanic crust or related fluids.

Currently there are limited Archean data available for either isotopic system. We hope that a broader dataset of robust isotopic measurements from a variety of Archean mantle-derived rocks will be established over time, similar to what is available for modern mantle-derived rocks. Until then, we believe our approach, which takes the measured $^{87}\text{Sr}/^{86}\text{Sr}$ from Archean anorthosites as an upper limit for the depleted mantle, accounts for potential mantle heterogeneity including possible mixing with altered oceanic crust.

Line-by-line comments:

L373: Lake Country or Lake County?

Correct, that should have been Lake County, thank you. Fixed.

L414: It looks like this isn't displaying properly here and elsewhere, should be HNO with 3 as the subscript, right?

Yes, thank you. Corrected.

L481: Double-check references, this one is formatted differently from the others.

Thanks, fixed.

Figures:

The figure captions for Figs 1 and 3 mention uncertainties +/- 2SE, however these are not apparent on the figure. Are the error bars smaller than the symbols? If so, it would be helpful to mention this.

Yes, that is the case here. Edited the figure captions to mention this.

For Fig 4d, there are some very light grey error bars that are difficult to see. The authors should darken these to make them more visible.

Thank you, the figure has been amended.

Supplementary material:

Supplementary Figure 7 – the caption mentions outliers, however there do not appear to be any.

That's correct. The figure caption has been fixed.

Supplementary Figure 9 – “Samples have MSWD >> 1, indicating all are disturbed to some degree” What about the three with MSWD < 2?

This is a good point. Added additional explanation to the caption.

Jillian Kendrick
Saint Mary's University
Halifax, Canada

References

- Bednarick AL, Bucholz CE, DePaolo DJ, Stolper DA (2024) Temporal covariation of island arc Sr isotopes and seawater chemistry over the past 2 billion years. *Proc Natl Acad Sci* 121:e2401832121. <https://doi.org/10.1073/pnas.2401832121>
- Caro G, Papanastassiou DA, Wasserburg GJ (2010) 40K–40Ca isotopic constraints on the oceanic calcium cycle. *Earth Planet Sci Lett* 296:124–132. <https://doi.org/10.1016/j.epsl.2010.05.001>
- Chen X, Zhou Y, Shields GA (2022) Progress towards an improved Precambrian seawater 87Sr/86Sr curve. *Earth-Sci Rev* 224:103869. <https://doi.org/10.1016/j.earscirev.2021.103869>
- Ivanic TJ, Nebel O, Jourdan F, et al (2015) Heterogeneously hydrated mantle beneath the late Archean Yilgarn Craton. *Lithos* 238:76–85. <https://doi.org/10.1016/j.lithos.2015.09.020>
- Kinny PD, Williams IS, Froude DO, et al (1988) Early archaean zircon ages from orthogneisses and anorthosites at Mount Narryer, Western Australia. *Precambrian Res* 38:325–341. [https://doi.org/10.1016/0301-9268\(88\)90031-9](https://doi.org/10.1016/0301-9268(88)90031-9)
- Lewis J, Luu T-H, Coath CD, et al (2022) Collision course; high-precision mass-independent and mass-dependent calcium isotope measurements using the prototype collision cell MC-ICPMS/MS, Proteus. *Chem Geol* 614:121185. <https://doi.org/10.1016/j.chemgeo.2022.121185>
- Linnebjerg B, Christiansen A, Maier WD, Szilas K (2025) Petrologic and Thermodynamic Constraints on the Petrogenesis of the Fiskenæsset Anorthosite Complex, SW Greenland: An Anhydrous Model for Archean Anorthosites. *J Petrol* 66:egaf027. <https://doi.org/10.1093/petrology/egaf027>
- Lowrey JR, Wyman DA, Ivanic TJ, et al (2024) Reappraising the petrogenesis of 2820–2738 Ma magmas of the western Youanmi Terrane, Western Australia. *Precambrian Res* 410:107456. <https://doi.org/10.1016/j.precamres.2024.107456>
- Myers JS (1988a) Oldest known terrestrial anorthosite at Mount Narryer, Western Australia. *Precambrian Res* 38:309–323. [https://doi.org/10.1016/0301-9268\(88\)90030-7](https://doi.org/10.1016/0301-9268(88)90030-7)
- Myers JS (1988b) Early Archean Narryer Gneiss Complex, Yilgarn Craton, Western Australia. *Precambrian Res* 38:297–307. [https://doi.org/10.1016/0301-9268\(88\)90029-0](https://doi.org/10.1016/0301-9268(88)90029-0)
- Polat A, Appel PWU, Fryer B, et al (2009) Trace element systematics of the Neoproterozoic Fiskenæsset anorthosite complex and associated meta-volcanic rocks, SW Greenland: Evidence for a magmatic arc origin. *Precambrian Res* 175:87–115. <https://doi.org/10.1016/j.precamres.2009.09.002>
- Polat A, Longstaffe FJ (2014) A juvenile oceanic island arc origin for the Archean (ca. 2.97 Ga) Fiskenæsset anorthosite complex, southwestern Greenland: Evidence from oxygen isotopes. *Earth Planet Sci Lett* 396:252–266. <https://doi.org/10.1016/j.epsl.2014.04.012>
- Rowe ML, Kemp AIS (2020) Spinel, olivine, and pyroxene chemistry of the Eoarchaean Manfred Complex (Yilgarn Craton, Western Australia), with implications for the tectonic setting of Archean layered mafic intrusions and the stabilisation of continental nuclei. *Lithos* 356–357:105340. <https://doi.org/10.1016/j.lithos.2019.105340>

Sobolev AV, Hofmann AW, Kuzmin DV, et al (2007) The Amount of Recycled Crust in Sources of Mantle-Derived Melts. *Science* 316:412–417. <https://doi.org/10.1126/science.1138113>

Vervoort JD, Kemp AIS (2024) Isotope Evolution of the Depleted Mantle. *Annu Rev Earth Planet Sci.* <https://doi.org/10.1146/annurev-earth-031621-112052>

Wang Z-N, Wang Y, Sossi P, et al (2024) Calcium isotopic composition of the bulk silicate Earth: A komatiite perspective. *Lithos* 480–481:107638. <https://doi.org/10.1016/j.lithos.2024.107638>